# The Prognostic Value of Neutrophil-to-Lymphocyte Ratio and Platelet-to-Lymphocyte Ratio for Small Renal Cell Carcinomas after Image-Guided Cryoablation or Radio-Frequency Ablation

**DOI:** 10.3390/cancers15072187

**Published:** 2023-04-06

**Authors:** Aqua Asif, Vinson Wai-Shun Chan, Filzah Hanis Osman, Jasmine Sze-Ern Koe, Alexander Ng, Oliver Edward Burton, Jon Cartledge, Michael Kimuli, Naveen Vasudev, Christy Ralph, Satinder Jagdev, Selina Bhattarai, Jonathan Smith, James Lenton, Tze Min Wah

**Affiliations:** 1Royal Surrey NHS Foundation Trust, Surrey GU2 7XX, UK; 2Division of Surgery and Interventional Science, University College London, London WC1E 6BT, UK; 3Leeds Institute of Medical Research, University of Leeds, Leeds LS2 9JT, UK; 4Royal Derby Hospital, University Hospitals of Derby and Burton NHS Foundation Trust, Derby DE22 3NE, UK; 5School of Medicine, Faculty of Medicine and Health, University of Leeds, Leeds LS2 9JT, UK; 6Chesterfield Royal Hospital NHS Foundation Trust, Chesterfield S44 5BL, UK; 7Royal Free London NHS Foundation Trust, London NW3 2QG, UK; 8School of Medical Education, Faculty of Medical Sciences, Newcastle University, Newcastle upon Tyne NE1 7RU, UK; 9Department of Urology, St. James’s University Hospital, Leeds LS9 7TF, UK; 10Department of Medical Oncology, St. James’s University Hospital, Leeds LS9 7TF, UK; 11Department of Pathology, St. James’s University Hospital, Leeds LS9 7TF, UK; 12Department of Diagnostic and Interventional Radiology, Institute of Oncology, Leeds Teaching Hospitals Trust, St. James’s University Hospital, Leeds LS9 7TF, UK

**Keywords:** renal cell carcinoma, image-guided ablation, cryoablation, radio-frequency ablation, neutrophil-to-lymphocyte ratio, platelet-to-lymphocyte ratio

## Abstract

**Simple Summary:**

This study investigated the use of simple blood tests, the neutrophil-to-lymphocyte ratio and the platelet-to-lymphocyte ratio to predict the outcome of renal cancer after being treated using image-guided ablation. We found these blood tests to predict worsened survival rates from cancer and risk of metastasis. We also found patients with a raised platelet-to-lymphocyte ratio to have significantly worsened kidney function post-operatively.

**Abstract:**

There is a lack of cheap and effective biomarkers for the prediction of renal cancer outcomes post-image-guided ablation. This is a retrospective study of patients with localised small renal cell cancer (T1a or T1b) undergoing cryoablation or radiofrequency ablation (RFA) at our institution from 2003 to 2016. A total of 203 patients were included in the analysis. In the multivariable analysis, patients with raised neutrophil-to-lymphocyte ratio (NLR) and platelet-to-lymphocyte ratio (PLR) pre-operatively, post-operatively and peri-operatively are associated with significantly worsened cancer-specific survival, overall survival and metastasis-free survival. Furthermore, an increased PLR pre-operatively is also associated with increased odds of a larger than 25% drop in renal function post-operatively. In conclusion, NLR and PLR are effective prognostic factors in predicting oncological outcomes and peri-operative outcomes; however, larger external datasets should be used to validate the findings prior to clinical application.

## 1. Introduction

Renal cancer carcinoma (RCC) is one of the most common cancers globally. Indeed, RCCs account for more than 90% of all malignant tumours of adult renal parenchyma. Due to local recurrence, limited drug response, and metastasis, long-term outcomes for RCCs are not currently optimised.

Patients with small renal masses are often managed by partial or radical nephrectomy. However, the complication rate of partial nephrectomy (PN) is high (~20%) and is associated with a significant gradual decline in renal function. An emerging adoption of image-guided ablation (IGA) for the treatment of small renal masses has arisen due to lower complication rates, the potential ability to preserve renal function, and its minimally invasive nature. A recent systematic review and meta-analysis comparing the ablative therapies and partial nephrectomy showed comparable oncological outcomes, superior renal function preservation and superior complication rates in ablation patients in comparison to partial nephrectomy patients [1]. Similarly, a retrospective study with 10 years of experience also showed comparable oncological outcomes and superior renal function in patients undergoing IGA compared to PN [2]. In specifically selected patients, active surveillance may be a viable option [3]. With multiple treatment options for small renal masses, it is therefore important to use image-guided biopsies [4] and other prognostic factors to select patients for the treatments with optimal treatment outcomes.

Prognostic factors are key to assisting clinicians in creating management plans for their patients. Accordingly, inflammation is hypothesised to play a key role in carcinogenesis; chronic inflammation may favour tumour development by suppressing anti-tumour activity [5]. Thus, biomarkers that have roles in the systemic inflammatory response may be suitable as prognostic factors. Inflammation results in neutrophilia through the stimulation of the production and migration of neutrophils. This may suppress the cytolytic activity of other cells, such as lymphocytes. Additionally, platelets have been shown to be associated with angiogenesis and may promote metastasis.

Examples of biomarkers include the neutrophil-to-lymphocyte ratio (NLR) and the platelet-to-lymphocyte ratio (PLR). NLR is defined as the absolute neutrophil count divided by the absolute lymphocyte count; similarly, PLR is defined as the absolute platelet count divided by the absolute lymphocyte count. These markers are cheap to carry out, are readily available to test, and are routinely performed in clinical settings. Whilst NLR and PLR have been shown to be prognostic factors for RCC, they have not been investigated for patients post-cryoablation. Only NLR has been investigated for patients undergoing RFA (radiofrequency ablation) [5] and for curative surgery [6]. Thus, this is the first study investigating the relationship between NLR and PLR and the outcomes of image-guided cryoablation or RFA using 10-year outcomes.

## 2. Materials and Methods

### 2.1. Study Design

This is a retrospective analysis of a prospective database of a maintained registry of patients undergoing image-guided cryoablation (CRYO) or RFA from 2003 to 2016 at a single institute. All adults with cT1N0M0, as defined according to the American Joint Committee on Cancer staging manual [7], and biopsy-proven (before or during ablation) RCC were included in the study. The selection criteria for RFA and CRYO were previously outlined [8]. Patients with a previous history of cancer of the kidney, solitary kidney, and inherited RCC syndromes, such as VHL, were excluded from the analysis [9]. The primary outcome of the study is the diagnostic value of NLR and PLR for cancer-specific survival (CSS). Secondary outcomes include the diagnostic value of NLR and PLR for overall survival (OS), local-recurrence-free survival (LRFS), metastasis-free survival (MFS), major complications, measured using the Clavien–Dindo scale [10], and a larger than 25% drop in eGFR post-operatively.

### 2.2. Performance of Image-Guided Ablation

The performance of RFA was described in detail previously [8]. RFA was performed under CT guidance using pulsed RF currents delivered by an impendence-controlled 200-W generator (Boston Scientific, Boston, MA, USA) using varying sizes (3, 3.5 or 4 cm) of an umbrella-shaped multi-tines LeVeen CoAccess RFA needle electrode. All CRYO were also performed under CT guidance, using 4–8 cryoprobes, depending on the size and geometry of the tumour. Two cycles of freezing and thawing were performed. During the process, an iceball was formed, covering the whole tumour using the Joule–Thomson effect with Argon–Helium gas delivery from the cryoablation generator (Boston Scientific, Maple Grove, MN, USA).

### 2.3. Patient Follow-Up

Patient follow-up protocol post-IGA was previously described in detail [8]; all patients were followed up at 1, 3 and 6 months after the procedure before commencing on annual follow-up using MRI or CT. Local recurrence was defined as new area(s) of enhancement in the zone of ablation after at least one imaging study had shown a complete lack of enhancement in the treated area. Metastatic disease was defined as extra-renal disease on imaging confirmed or suspected to have originated from the kidney. Cancer-specific death was defined as any deaths from RCC.

### 2.4. Clinical Features, Variables, Covariates, and Data Acquisition

The prospectively maintained database was consulted for patient features, such as age or sex; treatment details, such as treatment modality and complications; and tumour details, such as histopathological details, the R.E.N.A.L. nephrometry score [11] and the size of the lesion. Estimated glomerular filtration rate (eGFR) was calculated using the CKD-EPI method [12]. A major complication is defined as any post-operative complications larger than or equal to 3 on the Clavien Dindo scale [10]. Neutrophil, platelet and lymphocyte ratios were measured pre-operatively and immediately post-operatively. The corresponding ratios were calculated by simple division, i.e., NLR is calculated by dividing the value of the neutrophil by the value of the lymphocyte. Change in NLR or PLR is measured by simple subtraction. Utilising the National Health Service (NHS) patient records, the patients were followed for their living status and cause of death until 25 January 2021.

### 2.5. Outcomes and Data Synthesis

Baseline characteristics were evaluated using the Kruskal–Wallis test and the Chi-Square test. An optimal cut-off for each outcome and variable was determined using the Cut-off Finder, a validated web application designed for this use [13]. For survival outcomes, a cut-off was derived using a log-rank test and a cox regression, where the cut-off is defined as the point with the most significant log-rank test split [13]. For other outcomes, a logistic regression model was performed, with the optimal cut-off defined as the point with the most significant Fisher’s exact test split [13]. From there forward, in survival outcomes, a univariable cox regression model was used to determine the relationship between the cut-off of variables and the outcome, and then a backward stepwise multivariable cox regression model was used to take baseline characteristics into account. The results are presented in the form of hazard ratios (HRs), 95% confidence intervals (95% CI) and a *p*-value. A significance level of 0.2 was indicated for removal from the model, while a significance level of 0.1 was used for addition to the model. Similarly, for categorical outcomes, a univariable logistic regression was used first and then followed by a backward stepwise multivariable logistic regression model. The results are presented using odds ratios (ORs), 95% CI and *p*-values. All analyses are two-tailed at a significance level of 0.05. All statistical analyses were performed on STATA/MP 16.0 (StataCorp, College Station, TX, USA).

## 3. Results

### 3.1. Baseline Characteristics

A total of 203 patients were included in the study. A total of 103 patients underwent CRYO, while 100 underwent RFA. The median (IQR) age of the cohort is 73 (65–78). The median (IQR) size of tumours is 3.05 cm (2.5–2.7), with a median R.E.N.A.L. nephrometry score of 7 (5–8). The pre-operative median baseline NLR and PLR is 2.82 (1.98–4.10) and 142.1 (111.0–198.2), respectively. Patients undergoing RFA tend to have a significantly smaller change in NLR pre- and post-operatively compared to those undergoing CRYO. The full baseline characteristics are outlined in Table 1.

### 3.2. Operative and Survival Outcomes

In total, 201 out of 203 patients (99.0%) achieved primary technical success and complete treatment response. All patients achieved overall technical success, with only two RFA patients requiring further treatment. At a median (IQR) follow-up duration of 93.5 months (70.8–130.8), a total of 46 deaths was observed (CRYO: 30, RFA: 16). Of those, three were RCC related (CRYO: 1, RFA: 2). Eleven local recurrences were observed (CRYO: 5, RFA: 6), and four metastatic events were observed (CRYO: 1, RFA: 3). The five-year and ten-year OS rates are 86.6% (95% CI: 81.0–90.6%) and 75.8% (95% CI: 68.6–81.5%), respectively. The five-year and ten-year CSS rates are both 98.4% (95% CI: 95.3–99.5%). The LRFS rates are 96.0% (95% CI: 91.7–98.1%) and 91.1% (95% CI: 83.5–95.3%) at five and ten years, respectively. The MFS rates are 97.9% (95% CI 94.4–99.2%) at both five and ten years. Full oncological outcomes, in comparison to partial nephrectomy, have been reported previously [2].

### 3.3. Prognostic Factor for Overall Survival

Table 2 outlines the univariable analysis of prognostic factors for overall survival, and the associated Kaplan–Meier graphs are shown in Figure 1. Older age, larger tumour size, T1b tumours, and high R.E.N.A.L. nephrometry scores were associated with significantly worsened overall survival. Both NLR and PLR were found to be predictors of survival outcome. Post-operative NLR, when considered continuously (HR 1.06 95% CI 1.02–1.10, *p* = 0.004) and with a cut-off of 9.63 (HR: 3.28 95% CI 1.82–5.93, *p* < 0.001), is associated with significantly worsened OS. Change in NLR, when considered continuously (HR 1.07 95% CI 1.03–1.11, *p* = 0.001) and with a cut-off of 5.62 (HR: 3.52 95% CI 1.97–6.33, *p* < 0.001), is also associated with significantly worsened OS. Pre-operative PLR of larger than 222.5 is also associated with significantly worsened OS (HR: 1.96, 95% CI 1.03–3.72, *p* = 0.037). However, a post-operative PLR of larger than 75.31 is associated with significantly improved OS (HR: 0.28, 95% CI 0.12–0.67, *p* = 0.002). Change in PLR is associated with worsened OS both continuously (HR 1.004, 95% CI 1.001–1.006, *p* = 0.010) and with a cut-off of larger than 50.61 (HR 3.34, 95% CI 1.81–5.80, *p* < 0.001). In multivariable analysis (Table 3), a pre-operative NLR of ≥3.795, post-operative NLR of ≥9.63, and peri-operative change in NLR of ≥5.62 were all associated with significantly worsened OS. Similarly, a pre-operative PLR of ≥222.5, post-operative PLR evaluated continuously, and a peri-operative change in PLR of ≥50.61 is associated with significantly worsened OS; however, a post-operative PLR of ≥75.31 is associated with significantly improved OS.

### 3.4. Cancer-Specific Survival

Table 3 outlines the results of the univariable analysis of various prognostic factors and CSS. Increasing tumour size was associated with significantly worsened CSS. An increase in post-operative NLR, measured continuously, or with a cut-off of ≥18.29 are both associated with significantly worsened CSS. Similarly, increased peri-operative change in NLR, measured continuously, or with a cut-off of ≥12.36, is associated with significantly worsened CSS. Both post-operative PLR and peri-operative change in PLR when measured continuously, and with cut-offs of ≥306.5 and ≥171.9 were associated with significantly worsened CSS, respectively. The association of NLR and PLR with CSS are outlined in the Kaplan–Meier graphs in Figure 2.

In the multivariable analysis (Table 4), all pre-operative NLR, post-operative NLR and peri-operative changes in NLR are associated with significantly worsened CSS both measured continuously and with cut-offs of ≥1.203, ≥18.29 and ≥12.36, respectively. Similarly, both post-operative PLR and peri-operative change PLR measured continuously and with cut-offs of ≥306.5 and ≥171.9, respectively, are associated with significantly worsened CSS.

### 3.5. Local Recurrence Free Survival

Table 5 outlines the univariable analysis of prognostic factors for LRFS. Increasing age, tumour size and R.E.N.A.L. nephrometry score are factors leading to significantly worsened LRFS. Furthermore, it was found that a post-operative NLR ≥ 5.38 or a peri-operative change in NLR of ≥8.42 is associated with significantly worsened LRFS. Similarly, when measuring continuously, increased peri-operative change in NLR is also associated with significantly worsened LRFS (Figure 3).

However, in multivariate analysis, all NLR and PLR factors were not found to be associated with LRFS (Table 6).

### 3.6. Metastasis-Free Survival

Table 7 outlines the prognostic factors associated with MFS. Post-operative NLR, when measured with a cut-off of ≥15.26 is associated with significantly worsened MFS. Similarly, increased peri-operative change in NLR and PLR are both associated with significantly worsened MFS when measured continuously or with a cut-off of ≥12.36 or ≥171.9, respectively (Figure 4).

On multivariate analysis, a pre-operative PLR of ≥1.20 was found to have significantly better MFS. However, patients with a post-operative PLR (cut-off of ≥127.8) or increased peri-operative change in PLR (measured continuously or by a cut-off of ≥171.9) is associated with significantly worsened MFS (Table 8).

### 3.7. Major Post-Operative Complications

A total of eight major (Clavien–Dindo grade ≥ 3) post-operative complications were observed. Univariable analysis (Table 9) suggested only increasing R.E.N.A.L. nephrometry score and a pre-operative PLR of ≥115.3 to be associated with a significantly increased risk of major complications. However, NLR and PLR were not found to be significantly associated with major complications in multivariate analysis (Table 10).

### 3.8. Larger Than 25% Reduction in Renal Function (eGFR)

Table 11 outlines the prognostic factors for a larger than 25% reduction in renal function post-operatively. Increasing age, use of cryoablation over radio-frequency ablation, and a pre-operative PLR of <118.0 are associated with a significantly increased risk of a larger than 25% reduction in renal function post-operatively. In multivariable analysis (Table 12), a pre-operative PLR of <118 is confirmed to have a significantly higher risk of a larger than 25% reduction in eGFR.

## 4. Discussion

We have found that, in general, increased post-operative or peri-operative change in NLR and PLR are significant negative predictors of overall survival, cancer-specific survival and metastasis-free survival, which is in agreement with previous studies [5,6,14,15,16,17,18,19,20,21,22,23,24] in this cohort of patients with localised small renal masses who have undergone image-guided ablation. Interestingly, we have also found PLR to be a significant predictor of a significant decline in renal function. This is replicated in other disease groups and organs, notably in metastatic renal cancer and hepatocellular carcinoma [25,26,27,28].

Tumour progression and its initial development are associated with inflammation. Recently, there has been substantial growth in the evidence demonstrating that indeed inflammatory markers could represent prognostic factors for numerous cancer types, including RCCs [5,6,14,15,16,17,18,19,20,21,22,23,24]. In cancer-associated systemic inflammation, distributions of inflammatory cells shift; neutrophilia, thrombocytosis, and relative lymphocytopenia are observed, thus serving as potential biomarkers for cancer outcomes. Accordingly, there is a link between poorer post-operative survival and pre-operative systematic inflammatory response, which we have shown in our study; this is reflected where high NLR represented significant neutrophilia and where high PLR represented both neutrophilia and lymphocytopenia, suggesting NLR and PLR are markers for cancer-associated systemic inflammation.

Firstly, neutrophilia can be caused by tumours directly and by associated inflammation. For example, tumours can secrete various cytokines and growth factors, such as granulocyte colony-stimulating factor, interleukin-17, interleukin-1, interleukin-8 and granulocyte-macrophage colony-stimulating factor, which all stimulate neutrophil production [29]. This, once again, suggests the relation between high NLR (high neutrophil count in comparison to lymphocyte count) and cancer-associated inflammation.

Conversely, tumours can also produce tumour necrosis factor-alpha, interferon-gamma, interleukin-10 and interleukin-12, which may *decrease* lymphocytic activity as they are *immunosuppressive* molecules [30]. Lymphocytes tend to reflect antitumoral features, i.e., cell-mediated immunity [31]. Thus, high NLR (low lymphocyte count in comparison to normal or high neutrophil count) values may promote aggressive tumour progression, inhibit anti-tumour immune responses and negatively allude to poor survival. High NLR values could be biomarkers for both local and systemic inflammation, creating favourable environments for tumour development. Sejima et al. [32] observed a relationship between the NLR value and the activity levels of the immune system, as measured by the rate of the surface receptor, *Fas* ligand. The Fas ligand is involved in cell apoptosis through the action of cytotoxic T cells. Low NLR (normal or high lymphocyte count in comparison to normal or low neutrophil count) was found to be associated with a high expression of *Fas* ligand, relating positively to improved overall survival [32].

Finally, tumours may also release thrombopoietin and interleukin-6—which stimulate thrombocytosis through various signalling pathways. Interleukin-6, indeed, induces neutrophil proliferation and platelet formation due to stimulating the differentiation of megakaryocytes [33]. Moreover, growth and survival of cancer cells, along with the formation of new blood vessels for supplying oxygen and nutrients, are further reinforced by the production of platelet-derived growth factor and vascular endothelial growth factor (VEG-F) from cancer cells [34]. Interestingly, in the majority of cases for the most common renal cancer subtype (ccRCC), the tumour suppressor gene *Von Hippel-Lindau (VHL)* is altered or inactivated, causing a loss of VHL [29]. This, in turn, leads to the downstream production of increased levels of VEG-F-mediated angiogenesis. We know that VEG-F is produced and released by tumour cells, neutrophils, monocytes, and platelets, and that this molecule impairs the immune system by reducing dendritic cells and lymphocytic activity [35]. Indeed, elevated PLR has been previously shown to be associated with reduced disease-free and overall survival rates in renal cancer [24]. Other studies have shown links between elevated PLR and shorter survival rates in RCC [23,36]. Furthermore, another group of investigators found high PLR in bladder cancer patients to be associated with high grade, high T stage, and larger tumour size [37]. Combining the effect of thrombocytosis of thrombopoietin and interleukin-6 released by tumour cells, raised platelet counts (opposed to low or normal lymphocyte count) leads to increased PLR and hence worsened oncological outcome in patients as a negative predictor.

Interestingly, we found that a post-operative PLR of ≥75.31 was associated with a significantly improved OS. This finding may be attributed to the balance between inflammation and immune response in the tumour microenvironment. Lower post-operative PLR values, which indicate a reduced platelet count relative to lymphocytes, suggest a less favourable environment for tumour progression and a more robust anti-tumour immune response. Additionally, a lower PLR may reflect a reduced inflammatory response, a more favourable response to image-guided ablation, and a tumour microenvironment less conducive to cancer growth and metastasis. However, further research is needed to confirm and elucidate the exact mechanisms involved in this association.

The identification of higher-risk patients based on elevated NLR and PLR values can help clinicians make more informed decisions on treatment strategies, patient follow-up, and monitoring, ultimately leading to improved patient outcomes and more efficient resource allocation in healthcare settings. By incorporating NLR and PLR measurements into routine clinical practice, healthcare providers can utilise these biomarkers in conjunction with other established prognostic factors to tailor personalised treatment plans and the risk-stratify patients more accurately.

Furthermore, our findings open the door to exploring potential therapeutic interventions for patients with elevated NLR and PLR, such as more aggressive treatments, adjuvant therapies, or closer follow-up schedules. This personalised approach to patient care underscores the importance of precision medicine in oncology and may result in better clinical outcomes for patients with localised RCC.

Thus, our results are supported by the above and the extant literature regarding links between cancer prognosis and neutrophilia, lymphocytopenia, and thrombocytosis and prove that these states can exist concurrently [38].

### 4.1. Limitations

This was a retrospective, single-institutional study with a small sample size but a long follow-up period. We did not examine other potential factors that may be important for cancer progression and survival within the confines of this study, including generalised cardiovascular disease, hypertension, smoking status and diabetes. We acknowledge that NLR and PLR are not specific disease markers and that the presence of active infections or concurrent inflammatory disease states may have affected our study results. Furthermore, due to the lack of the number of events, CRYO and RFA must be analysed as one modality; however, in an attempt to reduce the effect of this on our results, a multivariable analysis model was used to take into account the potential variance due to the difference in the two modalities. Furthermore, only NLR and PLR have been investigated, and it is important to also investigate other inflammatory markers such as CRP.

### 4.2. Implications

We suggest that the multidisciplinary team should utilise the NLR and PLR to aid in the treatment decisions of patients with RCCs.

## 5. Conclusions

Neutrophil-to-lymphocyte and platelet-to-lymphocyte ratios and their respective change pre- and post-operatively are predictors for overall survival, cancer-specific survival and renal function. Our results will allow clinicians to make more well-informed decisions with the same volume of information they were likely to already have for patients undergoing image-guided cryoablation or radiofrequency ablation.

## Figures and Tables

**Figure 1 cancers-15-02187-f001:**
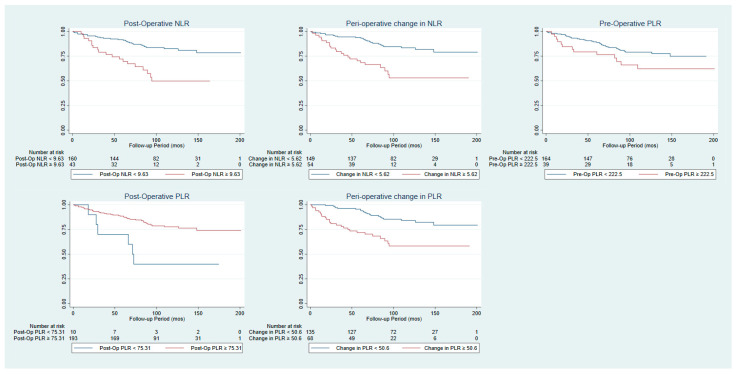
Kaplan–Meier curves comparing overall survival of patients with various cut-offs for post-operative NLR, peri-operative change in NLR, pre-operative PLR, post-operative PLR and peri-operative change in NLR.

**Figure 2 cancers-15-02187-f002:**
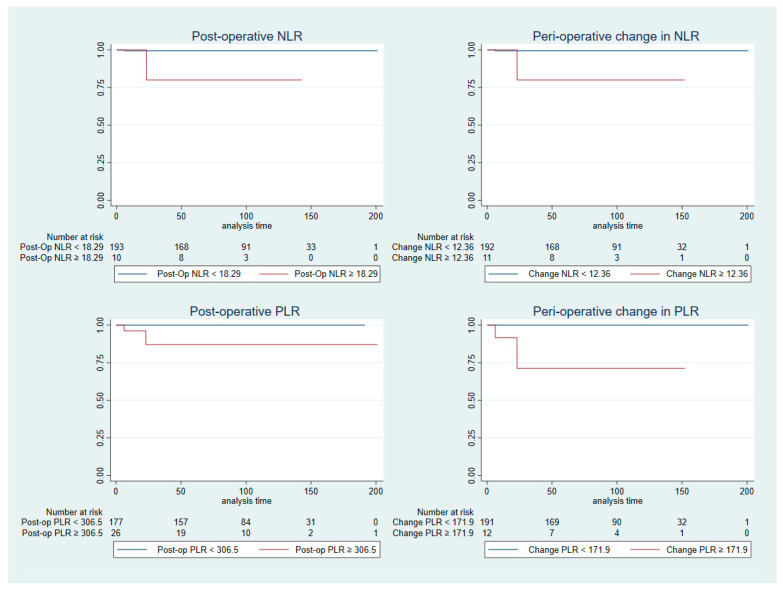
Kaplan–Meier graphs outlining the association between cancer-specific survival with PLR and NLR.

**Figure 3 cancers-15-02187-f003:**
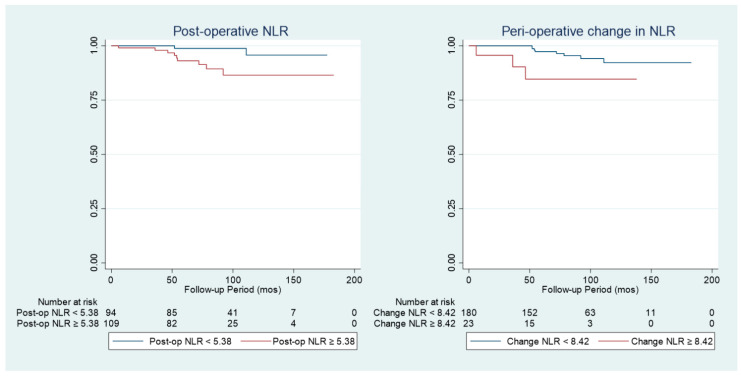
Kaplan–Meier graphs outlining the association of NLR with LRFS.

**Figure 4 cancers-15-02187-f004:**
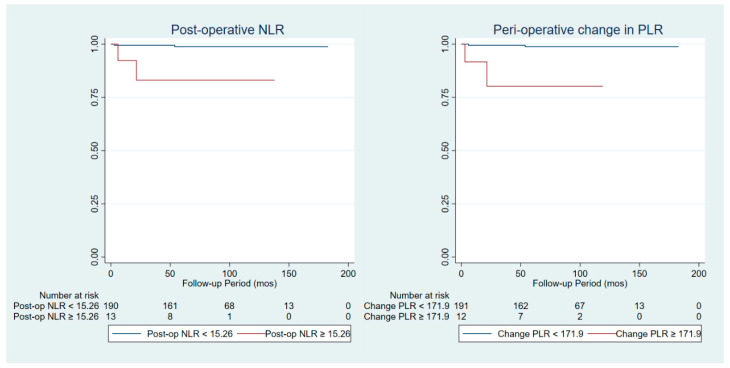
Kaplan–Meier graph outlining the relationship between NLR, PLR and metastasis-free survival.

**Table 1 cancers-15-02187-t001:** Baseline characteristics of included patients.

Modality	Cryoablation (n = 103)		RFA (n = 100)		Overall (n = 203)		*p*-Value (Chi-Squared)
Variable	Frequency	%	Frequency	%	Frequency	%	
Sex							
Male	64	62.1	65	35	129	63.5	0.672
Female	39	37.9	35	65	74	36.5	
Laterality							
Left	44	42.7	41	41.0	85	41.9	0.804
Right	59	57.28	59	59.0	118	58.1	
T stage							
T1a	72	69.9	87	87	159	78.3	0.003
T1b	31	30.1	13	13	44	21.7	
RCC Type							
Conventional	69	67.0	83	83.0	152	74.9	0.001
Papillary	8	7.8	5	5.1	13	6.4	
Oesinophil	2	1.9	6	6.1	8	3.9	
Chromophobe	24	23.3	5	5	29	14.3	
Fuhrman Grade							
Ungraded	15	14.6	12	12.0	27	13.3	0.569
1	21	20.4	23	23.0	44	21.7	
2	54	52.4	46	46.0	100	49.3	
3	11	10.7	17	17.0	28	13.8	
4	2	1.9	2	2.0	4	2.0	
	Median	IQR	Median	IQR	Median	IQR	*p*-value (Kruskal–Wallis)
Age	73	63–78	73	66–78	73	65–78	0.448
Tumour Size (cm)	3.3	2.6–4.1	3	2.5–3.6	3.05	2.5–3.7	0.035
R.E.N.A.L. Nephrometry Score	7	6–8	6	5–8	7	5–8	0.068
Baseline eGFR	71.7	55.0–87.4	84.7	62.1–103.2	78.4	56.8–95.3	0.005
Charlson Comorbidity Index	4	2–5	4	3–5	4	3–4	0.145
Baseline NLR	2.69	1.98–3.82	2.98	1.98–4.34	2.82	1.98–4.10	0.431
Post-op NLR	6.39	4.09–10.10	5.38	3.52–8.70	5.69	3.79–9.08	0.059
Change in NLR	3.12	1.33–6.77	2.07	0.46–4.93	2.48	0.98–5.77	0.003
Baseline PLR	134.4	109.0–189.4	148.3	115.4–206.7	142.1	111.0–198.2	0.091
Post-op PLR	171.0	123.4–254.8	172.6	127.1–272.1	172.1	127.0–260.0	0.491
Change in PLR	31.6	−0.72–66.8	21.5	−14.8–64.1	26.4	−5.2–65.51	0.281

**Table 2 cancers-15-02187-t002:** Univariable analysis of prognostic factors and overall survival.

Factor	Category/Outcome	HR (95% CI)	*p*-Value
Sex	Male	Ref.	
	Female	0.64 (0.34–1.21)	0.171
Age	Continuous	1.07 (1.03–1.11)	**0.001**
Laterality	Left	Ref.	
	Right	1.11 (0.61–2.00)	0.729
T stage	T1a	Ref.	
	T1b	5.16 (2.89–9.20)	**<0.001**
RCC Type	Conventional	Ref.	
	Papillary	0.66 (0.16–2.73)	0.564
	Eosinophil	0.50 (0.07–3.66)	0.496
	Chromophobe	0.70 (0.27–1.78)	0.454
Fuhrman Grade			
	1	Ref.	
	2	1.14 (0.53–2.46)	0.741
	3	0.77 (0.26–2.29)	0.632
	4	2.44 (0.53–11.3)	0.254
Treatment Modality	CRYO	Ref.	
	RFA	0.45 (0.24–0.84)	**0.012**
Tumour size (cm)	Continuous	2.03 (1.54–2.67)	**<0.001**
R.E.N.A.L. Nephrometry Score	Continuous	1.21 (1.03–1.43)	**0.024**
Pre-operative eGFR	Continuous	0.98 (0.97–0.99)	**<0.001**
Baseline NLR	Continuous	0.99 (0.87–1.12)	0.836
	<3.795	Ref.	
	>3.795	1.65 (0.91–2.98)	0.096
Post-op NLR	Continuous	1.06 (1.02–1.10)	**0.004**
	<9.63	Ref.	
	>9.63	3.28 (1.82–5.93)	**<0.001**
Change in NLR	Continuous	1.07 (1.03–1.11)	**0.001**
	<5.62	Ref.	
	>5.62	3.52 (1.97–6.33)	**<0.001**
Baseline PLR	Continuous	1.00 (0.997–1.002)	0.860
	<222.5	Ref.	
	>222.5	1.96 (1.03–3.72)	**0.037**
Post-op PLR	Continuous	1.00 (0.9997–1.004)	0.084
	<75.31	Ref.	
	>75.31	0.28 (0.12–0.67)	**0.002**
Change in PLR	Continuous	1.004 (1.001–1.006)	**0.010**
	<50.61	Ref.	
	>50.61	3.24 (1.81–5.80)	**<0.001**

Bold value indicates statistical significance (*p* < 0.05).

**Table 3 cancers-15-02187-t003:** Multivariate analysis of NLR and PLR for overall survival.

Factor	Category/Outcome	HR (95% CI)	*p*-Value
Baseline NLR	Continuous	1.06 (0.93–1.21)	0.353
	<3.795	Ref.	
	>3.795	2.16 (1.12–4.17)	**0.021**
Post-op NLR	Continuous	1.03 (0.99–1.08)	0.127
	<9.63	Ref.	
	>9.63	2.43 (1.27–4.66)	**0.007**
Change in NLR	Continuous	1.04 (0.99–1.09)	0.101
	<5.62	Ref.	
	>5.62	2.65 (1.38–5.06)	**0.003**
Baseline PLR	Continuous	1.00 (0.999–1.01)	0.177
	<222.5	Ref.	
	>222.5	4.18 (1.94–9.02)	**<0.001**
Post-op PLR	Continuous	1.00 (1.00–1.01)	**0.046**
	<75.31	Ref.	
	>75.31	0.34 (0.13–0.91)	**0.033**
Change in PLR	Continuous	1.00 (1.00–1.01)	0.170
	<50.61	Ref.	
	>50.61	2.84 (1.47–5.46)	**0.002**

Bold value indicates statistical significance (*p* < 0.05).

**Table 4 cancers-15-02187-t004:** Multivariable analysis of NLR and PLR for cancer-specific survival.

Factor	Category/Outcome	HR (95% CI)	*p*-Value
Baseline NLR	Continuous	0.43 (0.11–1.69)	0.228
	<1.203	Ref.	
	>1.203	0.03 (0.00–0.64)	**0.025**
Post-op NLR	Continuous	1.19 (1.06–1.33)	**0.002**
	<18.29	Ref.	
	>18.29	50.10 (3.85–651–1)	**0.003**
Change in NLR	Continuous	1.26 (1.05–1.50)	**0.010**
	<12.36	Ref.	
	>12.36	34.20 (2.71–431.95)	**0.006**
Baseline PLR	Continuous	1.01 (0.99–1.03)	0.305
	<161.7	Ref.	
	>161.7	302.8 (0.63–144577.9)	0.069
Post-op PLR	Continuous	1.01 (1.00–1.01)	**0.008**
	<306.5	Ref.	
	>306.5	Infinity (0-inf)	**<0.001**
Change in PLR	Continuous	1.01 (1.00–1.02)	**0.001**
	<171.9	Ref.	
	>171.9	Infinity (0-inf)	**<0.001**

Bold value indicates statistical significance (*p* < 0.05).

**Table 5 cancers-15-02187-t005:** Univariable analysis of prognostic factors associated with local-recurrence free survival.

Factor	Category/Outcome	HR (95% CI)	*p*-Value
Sex	Male	Ref.	
	Female	0.93 (0.27–3.18)	0.909
Age	Continuous	1.12 (1.03–1.23)	**0.011**
Laterality	Left	Ref.	
	Right	1.29 (0.38–4.40)	0.688
T stage	T1a	Ref.	
	T1b	2.74 (0.80–9.38)	0.109
RCC Type	Conventional	Ref.	
	Papillary	Inestimable	
	Eosinophil	Inestimable	
	Chromophobe	0.47 (0.06–3.71)	0.477
Fuhrman Grade			
	1	Ref.	
	2	3.57 (0.45–28.6)	0.230
	3	2.65 (0.24–29.3)	0.428
	4	Inestimable	
Treatment Modality	CRYO	Ref.	
	RFA	1.06 (0.32–3.55)	0.925
Tumour size (cm)	Continuous	1.97 (1.12–3.48)	**0.020**
R.E.N.A.L. Nephrometry Score	Continuous	1.51 (1.05–2.16)	**0.026**
Pre-operative eGFR	Continuous	0.99 (0.97–1.01)	0.313
Baseline NLR	Continuous	0.96 (0.71–1.29)	0.773
	<2.666	Ref.	
	>2.666	2.43 (0.64–9.16)	0.18
Post-op NLR	Continuous	1.08 (1.00–1.16)	0.060
	<5.38	Ref.	
	>5.38	5.13 (1.1–23.82)	**0.02**
Change in NLR	Continuous	1.09 (1.01–1.19)	**0.028**
	<8.416	Ref.	
	>8.416	4.91 (1.27–18.96)	**0.011**
Baseline PLR	Continuous	1.00 (0.99–1.00)	0.656
	<161.7	Ref.	
	>161.7	2.86 (0.84–9.8)	0.08
Post-op PLR	Continuous	1.00 (0.98–1.01)	0.630
	<294.4	Ref.	
	>294.4	2.95 (0.78–11.18)	0.096
Change in PLR	Continuous	1.00 (1.00–1.01)	0.215
	<140.2	Ref.	
	>140.2	3.33 (0.72–15.51)	0.100

Bold value indicates statistical significance (*p* < 0.05).

**Table 6 cancers-15-02187-t006:** Multivariable analysis of NLR and PLR for local-recurrence free survival.

Factor	Category/Outcome	HR (95% CI)	*p*-Value
Baseline NLR	Continuous	0.96 (0.68–1.37)	0.827
	<2.666	Ref.	
	>2.666	2.40 (0.59–9.73)	0.219
Post-op NLR	Continuous	1.05 (0.97–1.14)	0.258
	<5.38	Ref.	
	>5.38	3.52 (0.73–16.91)	0.115
Change in NLR	Continuous	1.06 (0.97–1.15)	0.218
	<8.416	Ref.	
	>8.416	2.76 (0.67–11.39)	0.159
Baseline PLR	Continuous	0.98 (0.99–1.01)	0.569
	<161.7	Ref.	
	>161.7	2.43 (0.67–8.80)	0.176
Post-op PLR	Continuous	1.00 (1.00–1.01)	0.832
	<294.4	Ref.	
	>294.4	1.94 (0.48–7.78)	0.351
Change in PLR	Continuous	1.00 (0.97–1.01)	0.427
	<140.2	Ref.	
	>140.2	2.11 (0.43–10.28)	0.356

Bold value indicates statistical significance (*p* < 0.05).

**Table 7 cancers-15-02187-t007:** The association of various prognostic factors with metastasis-free survival.

	Category/Outcome	HR (95% CI)	*p*-Value
Sex	Male	Ref.	
	Female	1.70 (0.24–12.05)	0.597
Age	Continuous	0.98 (0.89–1.07)	0.636
Laterality	Left	Ref.	
	Right	0.72 (0.10–5.09)	0.739
T stage	T1a	Ref.	
	T1b	3.93 (0.55–27.9)	0.171
RCC Type	Conventional	Ref.	
	Papillary	Inestimable	
	Eosinophil	Inestimable	
	Chromophobe	Inestimable	
Fuhrman Grade			
	1	Ref.	
	2	Inestimable	
	3	Inestimable	
	4	Inestimable	
Treatment Modality	CRYO	Ref.	
	RFA	3.21 (0.33–30.94)	0.312
Tumour size (cm)	Continuous	1.97 (0.72–5.39)	0.186
R.E.N.A.L. Nephrometry Score	Continuous	1.25 (0.67–2.34)	0.490
Pre-operative eGFR	Continuous	1.01 (0.39–1.04)	0.386
Baseline NLR	Continuous	0.88 (0.50–1.58)	0.668
	<1.203	Ref.	
	>1.203	0.15 (0.02–1.41)	0.054
Post-op NLR	Continuous	1.10 (0.99–1.21)	0.068
	<15.26	Ref.	
	>15.26	16.34 (2.29–116.52)	**<0.001**
Change in NLR	Continuous	1.12 (1.01–1.24)	**0.034**
	<12.36	Ref.	
	>12.36	20.51 (2.88–145.85)	**<0.001**
Baseline PLR	Continuous	1.00 (0.98–1.01)	0.701
	<127.8	Ref.	
	>127.8	Infinity (0–infinity)	0.11
Post-op PLR	Continuous	1.00 (1.00–1.01)	0.229
	<306.5	Ref.	
	>306.5	7.71 (1.08–54.9)	0.016
Change in PLR	Continuous	1.00 (1.00–1.01)	**0.035**
	<171.9	Ref.	
	>171.9	21.56 (3.01–154.33)	**<0.001**

Bold value indicates statistical significance (*p* < 0.05).

**Table 8 cancers-15-02187-t008:** Multivariable analysis of association between NLR, PLR and metastasis-free survival.

Factor	Category/Outcome	HR (95% CI)	*p*-Value
Baseline NLR	Continuous	0.83 (0.37–1.87)	0.649
	<1.203	Ref.	
	>1.203	0.05 (0.003–0.745)	**0.030**
Post-op NLR	Continuous	1.08 (0.93–1.25)	0.326
	<15.26	Ref.	
	>15.26	5.79 (0.45–73.64)	0.176
Change in NLR	Continuous	1.09 (0.94–1.28)	0.250
	<12.36	Ref.	
	>12.36	6.13 (0.48–78.68)	0.164
Baseline PLR	Continuous	0.99 (0.97–1.01)	0.513
	<127.8	Ref	
	>127.8	Inestimatable	
Post-op PLR	Continuous	1.00 (1.00–1.01)	0.195
	<306.5	Ref.	
	>306.5	33.9 (1.95–590.32)	0.016
Change in PLR	Continuous	1.01 (1.00–1.02)	**0.028**
	<171.9	Ref.	
	>171.9	71.6 (4.47–1144.61)	**0.003**

Bold value indicates statistical significance (*p* < 0.05).

**Table 9 cancers-15-02187-t009:** Univariable analysis of prognostics factors for major post-operative complications.

Factor	Category/Outcome	OR (95% CI)	*p*-Value
Sex	Male	Ref.	
	Female	0.24 (0.29–1.98)	0.184
Age	Continuous	1.04 (0.95–1.13)	0.371
Laterality	Left	Ref.	
	Right	0.71 (0.17–2.92)	0.636
T stage	T1a	Ref.	
	T1b	2.25 (0.52–9.82)	0.279
RCC Type	Conventional	Ref.	
	Papillary	Inestimable	
	Eosinophil	Inestimable	
	Chromophobe	0.74 (0.09–6.25)	0.782
Fuhrman Grade			
	1	Ref.	
	2	1.66 (0.19–14.41)	0.646
	3	Inestimable	
	4	8.67 (0.42–177.31)	0.161
Treatment Modality	CRYO	Ref.	
	RFA	1.75 (0.41–7.54)	0.450
Tumour size (cm)	Continuous	1.88 (9.98–3.61)	0.057
R.E.N.A.L. Nephrometry Score	Continuous	1.65 (1.07–2.53)	0.023
Pre-operative eGFR	Continuous	1.00 (0.98–1.02)	0.976
Baseline NLR	Continuous	0.71 (0.40–1.25)	**0.232**
	<2.3	Ref.	
	>2.3	0.33 (0.76–1.42)	0.135
Baseline PLR	Continuous	0.99 (0.98–1.00)	0.144
	<115.3	Ref.	
	>115.3	0.21 (0.05–0.92)	**0.038**

Bold value indicates statistical significance (*p* < 0.05).

**Table 10 cancers-15-02187-t010:** Multivariable analysis of association between pre-operative NLR and PLR with major complications.

Factor	Category/Outcome	HR (95% CI)	*p*-Value
Baseline NLR	Continuous	0.66 (0.31–1.38)	0.270
	<2.30	Ref.	
	>2.30	0.18 (0.02–1.31)	0.090
Baseline PLR	Continuous	0.99 (0.98–1.00)	0.267
	<115.3	Ref	
	>115.3	0.19 (0.03–1.13)	0.068

Bold value indicates statistical significance (*p* < 0.05).

**Table 11 cancers-15-02187-t011:** Univariable analysis of prognostic factors associated with a larger than 25% drop in eGFR post-operatively.

Factor	Category/Outcome	OR (95% CI)	*p*-Value
Sex	Male	Ref.	
	Female	0.51 (0.16–1.61)	0.255
Age	Continuous	1.10 (1.02–1.18)	**0.014**
Laterality	Left	Ref.	
	Right	1.81 (0.61–5.35)	0.282
T stage	T1a	Ref.	
	T1b	1.12 (0.35–3.63)	0.846
RCC Type	Conventional	Ref.	
	Papillary	Inestimatable	
	Eosinophil	4.73 (0.84–26.5)	0.077
	Chromophobe	2.96 (0.93–9.41)	0.066
Fuhrman Grade			
	1	Ref.	
	2	1.35 (0.35–5.25)	0.664
	3	0.51 (0.05–5.12)	0.564
	4	Inestimable	
Tumour size (cm)	Continuous	1.13 (0.69–1.85)	0.635
R.E.N.A.L. Nephrometry Score	Continuous	1.28 (0.97–1.69)	0.087
Treatment Modality	CRYO	Ref.	
	RFA	0.20 (0.05–0.71)	0.013
Baseline eGFR	Continuous	1.01 (0.99–1.03)	0.173
Baseline NLR	Continuous	0.87 (0.65–1.17)	0.354
	<5.23	Ref.	
	>5.23	0.35 (0.05–3.02)	0.364
Baseline PLR	Continuous	0.99 (0.99–1.00)	0.329
	<118.0	Ref.	
	>118.0	0.33 (0.12–0.89)	**0.029**

Bold value indicates statistical significance (*p* < 0.05).

**Table 12 cancers-15-02187-t012:** Multivariable analysis of association between pre-operative NLR and PLR with a larger than 25% drop in eGFR post-operatively.

Factor	Category/Outcome	HR (95% CI)	*p*-Value
Baseline NLR	Continuous	0.80 (0.53–1.21)	0.294
	<5.23	Ref.	
	>5.23	0.41 (0.04–4.20)	0.456
Baseline PLR	Continuous	0.99 (0.98–1.00)	0.208
	<118	Ref	
	>118	0.24 (0.08–0.83)	**0.025**

Bold value indicates statistical significance (*p* < 0.05).

## Data Availability

The data can be shared up on request.

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
