# Peer review of "The Prognostic Value of Neutrophil-to-Lymphocyte Ratio and Platelet-to-Lymphocyte Ratio for Small Renal Cell Carcinomas after Image-Guided Cryoablation or Radio-Frequency Ablation"

_cancers, 2023, doi:10.3390/cancers15072187_

Round 1

Reviewer 1 Report

Cancers-1975846-peer-review-v1

The Prognostic Value of Neutrophil-to-Lymphocyte Ratio and 2 Platelet-to-Lymphocyte Ratio for Small Renal Cell Carcinomas 3 (RCCs) After Image-Guided Cryoablation (CRYO) or Radio-4 Frequency Ablation (RFA)

This manuscript reposts a single center retrospective analysis of 203 patients with small stage T1a or T1b renal cell cancer who underwent cryoablation or radiofrequency ablation of renal tumor from 2003 – 2016.

The study concludes that high neutrophil to lymphocyte ratio (NLR) and platelet to lymphocyte ratio (PLR) pre-operatively, post-operatively and peri-operatively are associated with significantly worsened cancer-specific survival, overall survival and metastasis-free survival. Furthermore, an increased PLR pre-operatively is also associated with increased odds of a larger than 25% drop in renal function post-operatively.

Need grammatical editors.

Title

Please remove abbreviations such as RCC, Cryo, and RFA from the title?

Introduction

Ok

Method

-       Should add which guideline was used for categorization of complications.

-       Were biopsies down before or during ablation?

-       Which statistical analysis was utilized to determine the cutoff 2.3 for NLR and 115 for PLR?

Results

-       The authors should measure response to the treatment as well.

-       The number of patients with residual tumor or local recurrence disease is not presented? Should be per modality (Cryo vs RFA).

Discussion

-       Please omit “To our knowledge, this is the first study”.  Please focus on the study’s main findings in the first paragraph.

-       Page 15, lines 285-291: please bring this right after the prior paragraph, not as an independent paragraph.

-       Page 15, lines 300-302: please omit this statement.

-       I could not find any discussion around role of platelets and how PLR predicts the outcome.

References

Please cite references for other drugs and organs in discussion, such as:

https://pubmed.ncbi.nlm.nih.gov/34830607/

https://pubmed.ncbi.nlm.nih.gov/32424595/

https://pubmed.ncbi.nlm.nih.gov/34296630/

https://pubmed.ncbi.nlm.nih.gov/29454639/

Figures

Fine

Tables

Fine

Author Response

Reviewer comments

Author comments 

Title - Please remove abbreviations such as RCC, Cryo, and RFA from the title?

Thank you for the comment. This has been removed.

Methods - Should add which guideline was used for categorization of complications.

Thank you for the comment. All complications are classified by the Clavien-Dindo grade. This has been added.

Methods - Were biopsies down before or during ablation?

Thank you for the question. Our practice changed significantly over the study period. There is a mixture of both biopsies completed before or during ablation. This has been added to the manuscript.

Methods - Which statistical analysis was utilized to determine the cutoff 2.3 for NLR and 115 for PLR?

Thank you for the comment. This was done using the methods outlined in line 137 (Ref. 13). We have elaborated further the models used to determine a cut off. Essentially, for survival outcomes the log rank test and the cox regression model was used to determine a cut off, which is defined as the point with the most significant log-rank test split. For other outcomes, cutoffs are based on the significant of correlation with binary variable using a logistic regression, with the cut off defined as the point with the most significant fisher’s exact test split.  This is now elaborated further in the manuscript.

Results - The authors should measure response to the treatment as well.

Thank you for the suggestions. All patients in the series achieved overall technical success and 99% achieved primary technical success (line 161), unfortunately it was not possible to evaluate the role of NLR and PLR in predicting technical success and treatment response. However, we do agree this may be interesting and could potentially be replicated in a larger series.

Results - The number of patients with residual tumor or local recurrence disease is not presented? Should be per modality (Cryo vs RFA).

Thank you for the comment. The number of patients with local recurrence disease are outlined in line 165. And the residual tumour and technical success rate is outlined in line 161 We have now broken this down by modality.

Discussion - Please omit “To our knowledge, this is the first study”.  Please focus on the study’s main findings in the first paragraph.

Thank you for this comment. We have amended the first sentence of our discussion accordingly.

Discussion - Page 15, lines 285-291: please bring this right after the prior paragraph, not as an independent paragraph.

We have adjusted this accordingly, thank you.

Discussion - Page 15, lines 300-302: please omit this statement.

Thank you for this comment. As advised, we have removed this statement.

Discussion - I could not find any discussion around role of platelets and how PLR predicts the outcome.

Thank you. We agree with this comment. We have amended our Discussion extensively to include more about the role of thrombocytosis, and included more evidence for PLR.

References - Please cite references for other drugs and organs in discussion, such as:

https://pubmed.ncbi.nlm.nih.gov/34830607/

https://pubmed.ncbi.nlm.nih.gov/32424595/

https://pubmed.ncbi.nlm.nih.gov/34296630/

https://pubmed.ncbi.nlm.nih.gov/29454639/

Thank you for your suggestion. These have now been added.

Reviewer 2 Report

This study investigated the use of simple blood tests, neutrophil to lymphocyte ratio and platelet to lymphocyte ratio to predict the outcome of renal cancer after the therapy of using image-guided ablation. They found these blood tests to predict worsened survival rates from cancer and risk of metastasis.The research methods and contents are relatively rich and innovative, and the research results have certain clinical guiding significance which may assist in predicting prognosis from a routine, readily available, cheap set of biomarkers.However there are some problems in the article that need to be corrected or explained.

1. What criteria are used to determine the cut-off values of relevant indicators involved in the manuscript and whether they have certain decision-making value? Please explain this.

2. Cryoablation and radiofrequency ablation have different treatment principles, please explain whether performing the analysis together will make the analysis result unclear.

3. NLR and PLR are not specific disease markers. Whether the presence of active infections or inflammatory diseases in the study population will affect the study results and whether these patients need to be excluded?

4. Why other indicators closely associated with inflammation such as C-reactive protein were not investigated?

5. The discussion section has less relevant content for PLR which make the discussion content brief and please elaborate it further.

Author Response

Reviewer comments 

Authors' response

What criteria are used to determine the cut-off values of relevant indicators involved in the manuscript and whether they have certain decision-making value? Please explain this.

Thank you for your comment. A validated method (reference 15) was used to determine the cutoff values. Essentially, for survival outcomes were determined using the log-rank test and cox regression. Other outcomes, the logistic model is used to determine a cutoff. This is now further elaborated under the methods section.

Cryoablation and radiofrequency ablation have different treatment principles, please explain whether performing the analysis together will make the analysis result unclear.

Thank you for your comment. It is appreciated that cryoablation and RFA have different treatment principles, therefore, our multivariable analysis has included the treatment modality as a factor and all analysis would have been adjusted for RFA and cryoablation and hence will not affect the results.

NLR and PLR are not specific disease markers. Whether the presence of active infections or inflammatory diseases in the study population will affect the study results and whether these patients need to be excluded?

Thank you – we definitely agree with this comment. We have added this onto our limitations section.

Why other indicators closely associated with inflammation such as C-reactive protein were not investigated?

Thank you for your comment. These have not been investigated as they are not a routine test at our institution for pre-operative patients. NLR and PLR are also simpler blood tests that every patient will have as part of their full blood count panel pre-operatively. We agree these may play a role and further investigations should be done.

The discussion section has less relevant content for PLR which make the discussion content brief and please elaborate it further.

Thank you, we have amended our Discussion extensively to include details on PLR and thrombocytosis.

Reviewer 3

The paper is well written and is intersting. It could add new informations to the future management of RCC. I suggest simplifying tables and results, because there are many informations that could comphuse the readers.

Thank you for your comment. We appreciate the number of tables and results, however, all information are considered essential and it is good practice to report all clinically relevant outcomes of both univariate and multivariate models.

Reviewer 3 Report

The paper is well written and is intersting. It could add new informations to the future management of RCC. I suggest simplifying tables and results, because there are many informations that could comphuse the readers. 

Author Response

Reviewer 3

The paper is well written and is intersting. It could add new informations to the future management of RCC. I suggest simplifying tables and results, because there are many informations that could comphuse the readers.

Thank you for your comment. We appreciate the number of tables and results, however, all information are considered essential and it is good practice to report all clinically relevant outcomes of both univariate and multivariate models.

Round 2

Reviewer 1 Report

Introduction, Methods, Results

Are improved and OK

Discussion

The discussion is still very weak and needs rewriting.

Page 15, lines 270-272: In the first paragraph, I agree than change of 270 NLR and PLR are significant predictors of overall survival, But what type of predictors, positive or negative. The association should be described in a better way.

Page 15, 2nd paragraph of discussion: The authors have described association between tumor progression with inflammation. Can they relate and use this to explain how their result reflects inflammation.

Page 15, 3rd paragraph of discussion: Please omit “We will thus briefly discuss neutrophilia, thrombocytosis, and lymphocytopenia, in 285 the context of tumor activity.” This is not how discussion is written.

Page 15, 3rd paragraph of discussion: it is wonderful that the authors describe the role of neutrophilia here. But, they need to explain how NLR reflects this and how higher ration means higher rate of neutrophilia in realtion to lymphocytes…

Page 15, 4th paragraph of discussion: same thing here. The authors need to explain how NLR interpreted for lymphocyte (what it means/also if the relationship is positive or negative) and relation to tumor outcome.

Page 15, 5th paragraph: Please correlate PLR to thrombocytopenia and also go through the detail of PLR to prognosis, if increasing or decreasing PLR positive or negative predictor of response and outcome.

The authros have put some context in the discussion without trying to utilize them to explain their findings.

Page 6, limitation: please omit “It is important to note the limitations of our study.” It is redundant.

Page 6, lines 335-336: please omit “We must remain cognizant that our findings are an observation of a correlation in 335 one disease state.”

Page 6, line 342: did the authors mean “Neutrophil-to-lymphocyte and platelet-to-lymphocyte ratios” or “Changes in neutrophil-to-lymphocyte and platelet-to-lymphocyte ratios”?

Page 6, line 342: “can be used to predict” change to “are predictors”

Page 6, lines 343-344: please omit “Thus, these measures could 343 assist in predicting prognosis from a routine, readily available, cheap set of biomarkers.”

Page 6, line 345: please replace “Our results are important as they will” with “Our results will”

Author Response

Reviewer Comments 

Author’s response 

Reviewer 1 

Page 15, lines 270-272: In the first paragraph, I agree than change of 270 NLR and PLR are significant predictors of overall survival, But what type of predictors, positive or negative. The association should be described in a better way. 

Thank you for your comment. This has now been adjusted to negative predictors.  

Page 15, 2nd paragraph of discussion: The authors have described association between tumor progression with inflammation. Can they relate and use this to explain how their result reflects inflammation. 

Thank you for your comment. We have expanded on this, and this has now been explicitly explained in the paragraph.   

Page 15, 3rd paragraph of discussion: Please omit “We will thus briefly discuss neutrophilia, thrombocytosis, and lymphocytopenia, in 285 the context of tumor activity.” This is not how discussion is written. 

Thank you, this has been omitted.  

Page 15, 3rd paragraph of discussion: it is wonderful that the authors describe the role of neutrophilia here. But, they need to explain how NLR reflects this and how higher ration means higher rate of neutrophilia in realtion to lymphocytes… 

Thank you, this has been amended.  

Page 15, 4th paragraph of discussion: same thing here. The authors need to explain how NLR interpreted for lymphocyte (what it means/also if the relationship is positive or negative) and relation to tumor outcome. 

Thank you, this has been amended. 

Page 15, 5th paragraph: Please correlate PLR to thrombocytopenia and also go through the detail of PLR to prognosis, if increasing or decreasing PLR positive or negative predictor of response and outcome. 

Thank you, this has been amended.  

The authros have put some context in the discussion without trying to utilize them to explain their findings. 

Thank you, this has been amended by addressing comments above.  

Page 6, limitation: please omit “It is important to note the limitations of our study.” It is redundant. 

Thank you, this has been removed.  

Page 6, lines 335-336: please omit “We must remain cognizant that our findings are an observation of a correlation in 335 one disease state.” 

Thank you, this has been removed.  

Page 6, line 342: did the authors mean “Neutrophil-to-lymphocyte and platelet-to-lymphocyte ratios” or “Changes in neutrophil-to-lymphocyte and platelet-to-lymphocyte ratios”? 

Thank you for this - this has been clarified to reflect all four scenarios.  

Page 6, line 342: “can be used to predict” change to “are predictors” 

Thank you, this has been adjusted.  

Page 6, lines 343-344: please omit “Thus, these measures could 343 assist in predicting prognosis from a routine, readily available, cheap set of biomarkers.” 

Thank you, this has been adjusted.  

Page 6, line 345: please replace “Our results are important as they will” with “Our results will” 

Thank you, this has been amended.  

Reviewer 2 Report

1. The full name of CYRO should be shown when it first appeared in the manuscript.

2. Lines 66-69 of the manuscript mentioned”With varying management options for small renal masses, it is therefore crucial to utilise image-guided biopsies[4]and prognostics factors to safely select patients for treatments with optimal treatment outcomes.”Personally speaking, the meaning of this sentence is not clear.

3. Lines 70-71 of the manuscript refered to "Prognostic factors are key to assist clinicians in creating management plans for their patients". It is suggested to add relevant contents on the influence of NLR and PLR on prognosis and subsequent treatment and management in the discussion section.

4. Line 191-192 of the manuscript mentioned that "however a post-operative 191 PLR of ≥ 75.31 is associated with significantly improved OS." What is the possible reason?

Author Response

Reviewer 2 

1. The full name of CYRO should be shown when it first appeared in the manuscript. 

Thank you, this has been adjusted.  

Lines 66-69 of the manuscript mentioned”With varying management options for small renal masses, it is therefore crucial to utilise image-guided biopsies[4]and prognostics factors to safely select patients for treatments with optimal treatment outcomes.”Personally speaking, the meaning of this sentence is not clear. 

Thank you, this has been adjusted.  

Lines 70-71 of the manuscript refered to "Prognostic factors are key to assist clinicians in creating management plans for their patients". It is suggested to add relevant contents on the influence of NLR and PLR on prognosis and subsequent treatment and management in the discussion section. 

Thank you for this comment, we have added two paragraphs in our discussion section. 

Line 191-192 of the manuscript mentioned that "however a post-operative 191 PLR of ≥ 75.31 is associated with significantly improved OS." What is the possible reason? 

Thank you for this, we have added a paragraph in our discussion to explain this possible association. 

Round 3

Reviewer 1 Report

Well done with revisions